# Safety of Three-Dimensional versus Two-Dimensional Laparoscopic Hysterectomy during the COVID-19 Pandemic

**DOI:** 10.3390/ijerph192114163

**Published:** 2022-10-29

**Authors:** Dariusz Kowalczyk, Szymon Piątkowski, Maja Porażko, Aleksandra Woskowska, Klaudia Szewczyk, Katarzyna Brudniak, Mariusz Wójtowicz, Karolina Kowalczyk

**Affiliations:** 1Department of Anatomy, School of Medicine in Opole, University of Opole, 45-052 Opole, Poland; 2Students’ Scientific Association of Gynecology and Obstetrics, School of Medicine in Opole, University of Opole, 45-052 Opole, Poland; 3Department of Gynecological and Obstetrics Women’s and Child Health Center, Medical University of Silesia, 41-803 Zabrze, Poland; 4Department of Endocrinological Gynecology, Faculty of Medicine in Katowice, Medical University of Silesia, 40-752 Katowice, Poland

**Keywords:** COVID-19 pandemic, three-dimensional laparoscopy, two-dimensional laparoscopy, laparoscopic supracervical hysterectomy, LASH, total laparoscopic hysterectomy, TLH

## Abstract

Background: The COVID-19 pandemic has resulted in a significant decrease in the number of surgical procedures performed. Therefore, it is important to use surgical methods that carry the lowest possible risk of virus transmission between the patient and the operating theater staff. Aim: Safety evaluation of three-dimensional (3D) versus two-dimensional (2D) laparoscopic hysterectomy during the COVID-19 pandemic. Methods: 44 patients were assigned to a prospective case-control study. They were divided either to 3D (*n* = 22) or 2D laparoscopic hysterectomy (*n* = 22). Fourteen laparoscopic supracervical hysterectomies (LASH) and eight total laparoscopic hysterectomies (TLH) were performed in every group. The demographic data, operating time, change in patients’ hemoglobin level and other surgical outcomes were evaluated. Results: 3D laparoscopy was associated with a significantly shorter operating time than 2D. (3D vs. 2D LASH 70 ± 23 min vs. 90 ± 20 min, *p* = 0.0086; 3D vs. 2D TLH 72 ± 9 min vs. 85 ± 9 min, *p* = 0.0089). The 3D and 2D groups were not significantly different in terms of change in serum hemoglobin level and other surgical outcomes. Conclusions: Due to a shorter operating time, 3D laparoscopic hysterectomy seems to be a safer method both for both the surgeon and the patient. Regarding terms of possible virus transmission, it may be particularly considered the first-choice method during the COVID-19 pandemic.

## 1. Introduction

In early December 2019, the first case of SARS-CoV-2 was reported in Wuhan, China [1]. A rapidly spreading epidemic paralyzed healthcare all over the world, and in numerous places, there was a significant decline in the level of surgical activity [2,3]. Unfortunately, apart from the decrease in the number of elective surgeries, the number of urgent procedures has also decreased [2]. There are various reasons for this, one of which is certainly the desire to reduce the risk of SARS-CoV-2 virus transmission between operated patients and the operating theater staff. An important task for healthcare systems worldwide at the present seems to be the creation of procedures, enabling the safest possible performance of surgical procedures. The virus is transmitted from one person to another by airborne droplets or through contact with the surface on which droplets from the respiratory system of an infected person have settled [4]. It can also reach human mucous membranes through contact with the aerosol generated during surgical procedures [5]. Therefore, laparoscopic surgery seems to be safer than open surgery because of the possibility of filtering bioaerosol and smoke generated during the procedure [6].

Since surgical trauma during laparoscopy is smaller than during open surgery, the patient stays for a shorter time in the healthcare facility where the procedure is performed. This may have a major impact on nosocomial SARS-CoV-2-related infections. During laparoscopy, bioaerosol is formed in the abdominal cavity, which could potentially contain virions, so it is important to avoid the free release of pneumoperitoneum during desuflation. It is worth using a HEPA filter for this process, which retains particles smaller than 0.3 μm with an efficiency of 99.97% [7]. In the study of M. Zago et al., filters were used in the suction device to release the pneumoperitoneum and at each port. This procedure proved effective, and none of the healthcare professionals participating in the study developed the disease or obtained a positive test for SARS-CoV-2 [8]. Detailed recommendations for the management of laparoscopic surgery during the COVID-19 pandemic are presented in Table 1.

In the 1990s, endoscopic surgery spread and was used in various specialties. In the following years, imaging systems enabling three-dimensional visualization were developed. The 3D-imaging method was first used in gynecology in 1993 [10]. Three-dimensional laparoscopes have two cameras placed next to each other, which simulate the left and right eyes. The images from the cameras overlap each other to enable the generation of stereoscopy [11] (Figure 1). The surgeon, after putting on special glasses in which the right and left lenses transmit light of different polarization, can see an image of illusory depth (Figure 2). The objective of this study was to evaluate the safety of 3D versus 2D laparoscopic hysterectomy during the COVID-19 pandemic.

## 2. Materials and Methods

### 2.1. Patients

A prospective case-control study was conducted between January and August 2021 in the Department of Gynecology and Obstetrics in the Hospital in Nysa, Poland. Women scheduled for elective laparoscopic surgery LASH or TLH were enrolled in the study. The indication for LASH was symptomatic uterine fibroids, whereas the indication for TLH was atypical endometrial hyperplasia. Every woman had undergone endometrial sampling and Pap test screening before qualification to surgery. Women suspected of gynecological malignancy were excluded from the study. Patients undergoing laparoscopic hysterectomy were divided into two groups: 2D and 3D laparoscopy. On admission to the hospital, each patient was randomly assigned alternately to one of the groups. It was performed by the same physician according to the order of admission.

The primary outcome was operating time. It was measured from the laparoscopic entry using Hasson’s technique to skin closure at the last trocar incision site. Secondary outcomes were defined as surgery-related changes in hemoglobin values and the rate of complications. Hemoglobin values were measured on the day before the operation and on the first postoperative day. Hospital stay was registered. Complications were classified as follows:Conversion to abdominal hysterectomy.Intraoperative complications: transfusion-related hemorrhage, injury to the bladder, intestine, ureter, or vessels, and anesthesiological problems.Postoperative complications: infection or temperature above 38.8C, hematomas, revisions/secondary procedures, deep venous thrombosis, fistula formation, and wound-healing disturbances.

The study protocol was approved by the Nysa Hospital Ethics Board on 10 December 2020, Approval Code NHEB/0020/KB/10/12/20. All patients signed consent forms, and their confidentiality and anonymity were protected.

### 2.2. Uterine Morcellation 

Patients with presumed leiomyomas were qualified to LASH with the use of power morcellator according to the American College of Obstetricians and Gynecologists Committee Opinion (ACOG) [12]. ACOG recommends a minimally invasive approach for women undergoing hysterectomy for benign disease whenever feasible [13]. Although leiomyosarcoma is not reliably identifiable preoperatively, the evaluation included risk stratification and the appropriate use of imaging techniques. Subjects were examined for gynecological malignancies with Pap test screening and endometrial sampling. Only premenopausal women were qualified for morcellation in the study. The gynecologist and patients were engaged in shared decision making. Finally, after explaining the risks and benefits of morcellation, alternatives to it, as well as the risks and benefits of abdominal hysterectomy vs. LASH, patients signed informed consent for this procedure.

### 2.3. Laparoscopic Equipment

Three ports were placed typically in every patient. For the camera system, a 10 mm 3D system EinsteinVision^®^ 3.0 (Aesculap Chifa Sp. z o.o., Nowy Tomyśl, Polska) and a 10 mm 2D system Full HD CMOS (Aesculap Chifa, Nowy Tomyśl, Polska) were used. Identical laparoscopic instruments were used in both groups.

### 2.4. Laparoscopic Technique

To minimize the variability of surgical skills, the same gynecological surgeon (D. Kowalczyk), who has experience of performing > 200 laparoscopic hysterectomy procedures, performed all operations. 

The surgical technique for TLH and LASH has been followed, as described by Mueller et al. [14]. The patient was placed in the lower Trendelenburg position. In both surgeries, the SecuFix Uterus Manipulator (Richrd Wolf GmbH, Knittlingen, Germany) was used to elevate the vaginal fornices and enable uterine movements during the procedures. The manipulator was introduced into the uterine cervix and firmly screwed together using a threaded piece. In this way, the adapter that was subsequently introduced enclosed the vaginal part of the cervix. Initially, the round ligament was transected, followed by the tube and the proper ligament of the ovary after bipolar coagulation. If an adnexectomy was being conducted at the same time, the ureter was demonstrated retroperitoneally before the transection of the suspensory ligament of the ovary. The uterine vascular pedicle was coagulated with bipolar diathermy and cut. The procedure was conducted likewise on each side. 

During TLH, the vesical peritoneum was then incised. After this, the manipulator, and with it, the entire uterus, was elevated into the abdomen so that tension was created at the cervicovaginal junction. This resulted in the dislocation of the vaginal fornices (and the vaginal part of the uterine cervix) from the bladder and from the distal part of the ureter, as well as lateralization of the bladder pillars over the edge of the adapter. The superior fornix of the vagina was easily identified and was separated with the monopolar needle along the edge of the adapter in the vaginal part, directly at the uterine cervix. The sacrouterine ligaments were preserved. The manipulator, together with the uterus, were then withdrawn into the vagina and removed through it. The vagina was closed with a V-lock suture (Medtronic, Minneapolis, MN, USA).

In LASH surgery, the first step of the operating procedure was the same as that described above. Both the anterior and posterior leaves of the broad ligament were cut after bipolar coagulation. The uterine artery was transected after bipolar coagulation. The uterine body was separated from the cervix in the isthmic area of the uterus using a monopolar needle. Subsequently, the uterine manipulator was withdrawn, and the corpus uteri was removed after morcellation with an electric morcellator. The cervix was closed with a V-lock suture.

### 2.5. Statistical Analysis

For continuous variables, data are presented as mean ± standard deviation (SD) after verifying the normal distribution of the data. The baseline characteristics and surgical outcomes were compared between the two groups using student’s *t*-test for continuous variables, and the chi-square test or Fisher’s exact test for categorical variables, as appropriate. Statistical significance was set at *p* < 0.05. All the statistical analyses were conducted using STATISTICA 12.0 PL (StatSoft Polska, Krakow, Poland).

## 3. Results

During the study period, 44 patients underwent laparoscopic hysterectomy. Fourteen LASH and eight TLH procedures were performed in every group: 3D laparoscopy (*n* = 22) and 2D laparoscopy (*n* = 22). The demographic data of the study groups are presented in Table 2.

There were no significant differences between the groups regarding the age of patients who underwent LASH (56 ± 7.3 years in 2D vs. 58 ± 10 years in 3D group, *p* > 0.05). The mean BMI of 2D and 3D LASH patients also did not differ significantly (26.5 ± 2.3 vs. 25.8 ± 2.7 kg/m^2^). The average time of the LASH procedures in 3D imaging was shorter by 20 min than in 2D imaging (70 ± 23 vs. 90 ± 20 min, respectively, *p* = 0.0086). Change in the serum hemoglobin level was similar in both groups (1.4 ± 1.3 vs. 1.6 ± 1.6 g/dL; *p* > 0.05). One of the 2D LASH procedures required conversion to the open method. Other surgical outcomes, including postoperative complications and hospitalization length, were comparable between the groups.

In line with the above criteria, we compared 2D and 3D TLH procedures. Neither age nor BMI between the 2D and 3D groups differed significantly (Table 2). The operative time in the 3D group was significantly shorter than in the 2D group (72 ± 9 vs. 85 ± 9 min respectively; *p* = 0.0089). The rest of the surgical outcomes are presented in Table 2.

The surgeon emphasized, subjectively, having more precise movements with complex mechanisms and fewer uncontrolled movements in 3D imaging. Work comfort and hand–eye coordination were comparable in both methods. During the procedures in 3D projection, the surgeon and assistant surgeons did not experience fatigue or headache. 

## 4. Discussion

As shown in this study, 3D laparoscopy is a significantly faster method than 2D laparoscopy in LASH and TLH procedures. These findings are in line with numerous studies comparing 3D and 2D TLH [15,16,17]. Berlit et al. were concerned about the role of 3D hysterectomy in overweight and obese patients [18]. A significantly lower blood loss was revealed using 3D visualization in the LASH subgroups of the normal and overweight collectives. Three-dimensional laparoscopy was additionally associated with a significantly shorter duration of surgery in the TLH subgroup in overweight patients and a lower hemoglobin drop in the LASH subgroup of the obese. Interestingly, they found that an additional benefit of 3D laparoscopic hysterectomy was fewer trocar site incisions in all BMI groups. Similar results were presented in a meta-analysis, where, based on 31 studies, speed, precision, side effects and cognitive workload were assessed. Three-dimensional imaging improved the speed and reduced complications in laparoscopy [19]. A lower complication rate was also found in a study on three-dimensional laparoscopic cholecystectomy compared to the two-dimensional method [20]. A lower number of postoperative complications and a shorter procedure time were obtained by comparing the laparoscopic distal gastrectomy in 3D imaging compared to the 2D method [21]. The use of 3D imaging in laparoscopic appendectomy was associated with a reduction in the operative time compared to the 2D method. It is worth noting that the operators were young surgeons [22]. On the other hand, no surgical benefits were found in a randomized controlled trial concerning 3D versus 2D laparoscopic myomectomy [23]. A total of 64 patients with symptomatic uterine fibroids were randomly assigned to either the 3D (*n* = 32) group or the 2D group (*n* = 32). The 3D and 2D groups were not significantly different in terms of operative blood loss, change in serum hemoglobin levels (1.4 ± 1.6 g/dL vs. 1.6 ± 1.6 g/dL, *p* = 0.553), and operative time (77.4 ± 37.8 min vs. 82.4 ± 35.4 min, *p* = 0.344). Furthermore, no differences were observed between the groups regarding other surgical outcomes [23].

The learning curve and training time of young surgeons are also important in comparing these two methods. A randomized study comparing the ability to master laparoscopic skills by 40 beginners with the use of trainers indicated better efficiency and precision, thanks to 3D visualization [24]. Moreover, similar effects were achieved with this technique much faster than in the 2D technique [24,25,26].

Even for experienced laparoscopic surgeons, the flat, depthless image is problematic. The solution to this problem seems to be the use of optics, enabling the use of a three-dimensional image, which will improve perception during the procedure. A. Zwimpfer et al. showed that, regardless of the operator’s experience, the use of 3D imaging reduces the time, increases accuracy, and reduces the number of errors made during the task [27]. The depth perception obtained thanks to the use of 3D imaging increases the quality of visualization, which leads directly to the number of detected changes. In the analysis comparing the quality of imaging, out of 598 lesions of the endometriotic nature, 595 lesions (99.5%) were spotted using the 3D technique, and 474 lesions (79.3%) were spotted with the conventional technique. Three-dimensional laparoscopy also detected nearly twice as many peritoneal defects as in 2D imaging [28].

The little available data show that the risk of transmission of SARS-CoV-2 in a surgical healthcare unit with universal masking and appropriate hand hygiene is low [29]. In this cohort of healthcare workers, the antibody positivity rate was also low and consistent with community rates [30]. Nonetheless, as the COVID-19 pandemic continues, an important role is to reduce the risk of infection for both staff and patients. Laparoscopy can be safe with precautionary measures and shorten the patient’s hospital stay [31]. To ensure greater safety during laparoscopic procedures, it is worth using filters and avoiding the sudden and frequent insertion and removal of surgical instruments through the ports [32,33]. However, more research is needed on the infectivity of surgical aerosol and virus transmission [34]. 

Given the shorter operative time, lower blood loss and lower complication rate, we can presume that 3D laparoscopy is potentially a cost-effective method. Feng et al. did not observe differences in the total cost of laparoscopic radical cystectomy in the 3D versus 2D technique [35]. However, the authors of systematic reviews comparing these two modalities concluded that more investigations on bigger cohort sizes and unique 3D visual systems are needed to justify its cost effectiveness [19,36]. The next stage in the evolution of 3D laparoscopy is robotic surgery. Unfortunately, since their high costs, they will not be a reference method for the coming years. However, there are studies that show that similar results as in robotic surgery can be reached with the use of laparoscopy in 3D imaging [37,38]. According to the authors, 3D laparoscopy may gradually displace it due to its numerous advantages over the 2D method.

## 5. Conclusions

The decrease in the total number of surgical procedures underlines the need for systemic changes in the approach for undertaking surgical activity during the SARS-CoV-2 pandemic. Three-dimensional LASH and TLH can be considered safe and even the reference methods of laparoscopic hysterectomy due to the shorter procedure time. The use of the laparoscopic technique in 3D imaging can additionally reduce the risk of exposure of the medical staff to SARS-CoV-2 infection during the COVID-19 pandemic.

## Figures and Tables

**Figure 1 ijerph-19-14163-f001:**
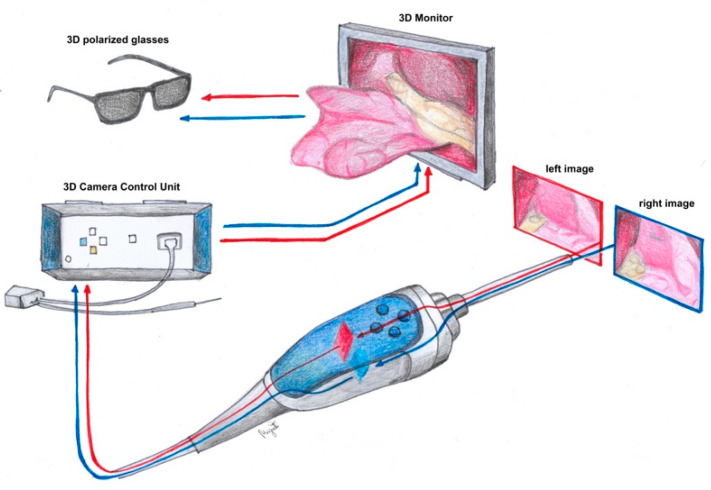
Diagram showing principles of 3D laparoscopy.

**Figure 2 ijerph-19-14163-f002:**
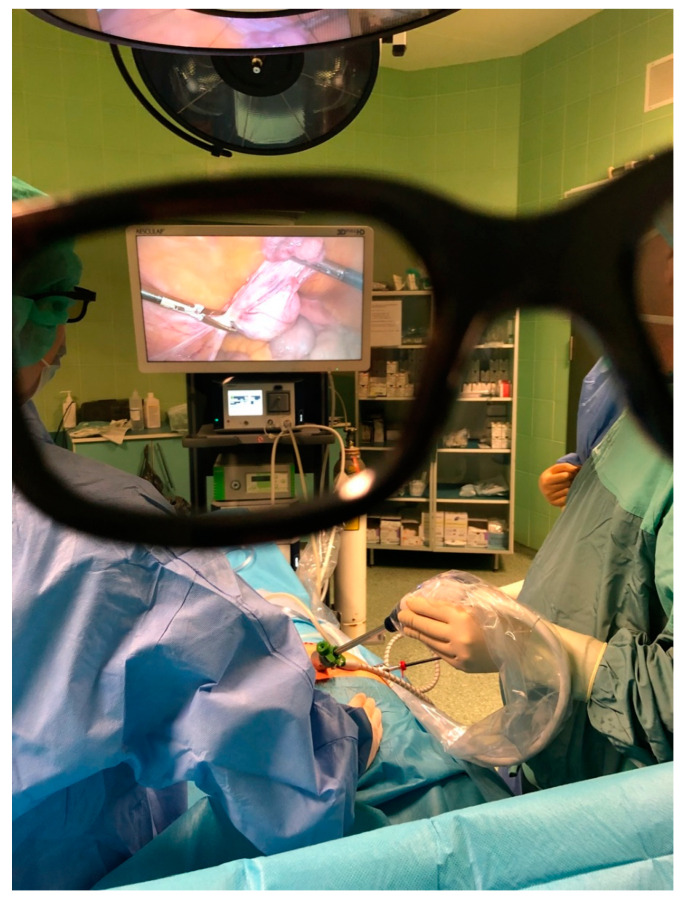
Final 3D image seen by the surgeon during laparoscopy.

**Table 1 ijerph-19-14163-t001:** Recommended procedures in laparoscopic surgery increasing the safety of medical personnel in the time of the COVID-19 pandemic [9].

The Number of People in the Operating Room Should Be Kept to a Minimum. Those Present in the Room Should Be Adequately Experienced to Minimize the Time of the Procedure.
When using anesthesia equipment, it is recommended to use HEPA filters to minimize the spread of exhaled infectious material by the patient.
Desuflation should be performed slowly, using HEPA filters to limit excess gas dispersion.
Tools used during procedures in COVID-19 patients should not be sterilized together with tools used in non-COVID-19 patients.
In many cases, postponing the procedure to the optimal time should be considered. In the case of endometrial cancer surgery, early surgery increases the number of vascular complications and infections and extends the length of hospital stay.
Before surgery, PCR tests should be performed, and a chest CT scan should be considered.
Avoid the spread of blood droplets in the operating room.
It is worthwhile to carefully secure the access ports for tools against gas leakage.
It should be remembered that any surgery is associated with immunosuppression, which may increase the risk of infection and the severe course of COVID-19 in the operated patient.

**Table 2 ijerph-19-14163-t002:** Demographic data and surgical outcomes.

	2D LASH (*n* = 14)	3D LASH (*n* = 14)	*p* Value
Age, year	56 ± 7.3	58 ± 10	*p* > 0.05
Body mass index, kg/m^2^	26.5 ± 2.3	25.8 ± 2.7	*p* > 0.05
Operative time, min	90 ± 20	70 ± 23	*p* = 0.0086
Change in serum hemoglobin, g/dL	1.4 ± 1.3	1.6 ± 1.6	*p* > 0.05
Conversion to open surgery	1	0	*p* > 0.05
Postoperative complications	0	0	*p* > 0.05
Length of hospital stay, days	3 ± 0.2	3 ± 0.3	*p* > 0.05
	**2D TLH** **(*n* = 8)**	**3D TLH** **(*n* = 8)**	***p* Value**
Age, year	62 ± 9.7	60 ± 9.2	*p* > 0.05
Body mass index, kg/m^2^	25.3 ± 3.2	25.7 ± 3.7	*p* > 0.05
Operative time, min	85 ± 9	72 ± 9	*p* = 0.0089
Change in serum hemoglobin, g/dL	1.2 ± 1.4	1.1 ± 1.3	*p* > 0.05
Conversion to open surgery	0	0	*p* > 0.05
Postoperative complications	0	0	*p* > 0.05
Length of hospital stay, days	3 ± 0.4	3 ± 0.3	*p* > 0.05

Values are presented as mean ± standard deviation (SD) after verifying the normal distribution of the data.

## Data Availability

Not applicable.

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
