# Peer review of "Safety of Three-Dimensional versus Two-Dimensional Laparoscopic Hysterectomy during the COVID-19 Pandemic"

_ijerph, 2022, doi:10.3390/ijerph192114163_

Round 1
Reviewer 1 Report
The authors raised an interesting topic regarding the search for more effective and thus safer operating methods.
The methodology is described in detail, however, the authors should provide the approval number of the bioethics committee, which is usually obtained in academic centers.
In Table 2, some of the results given do not agree with those described earlier in the text: regarding the age of LASH patients and hemoglobin changes. Please review and correct carefully.
In line 201 and 202 discussions, p=0.553 should be used instead of P=0.553 and p=0.344.
Conclusions:
It seems that the authors should include a sentence referring to the superiority of the 3D LASH and TLH method over 2D, without narrowing the benefits only for the times of the COVID-19 pandemic.
Reviewer 2 Report
The study "Safety of three-dimensional versus two-dimensional laparoscopic hysterectomy during the COVID-19 pandemic" by Kowalcsyck et al is a small randomised study intending to ascertain if laparoscopic hysterectomy using 3D technology is quicker than using 2D technology.
The introduction is appropriate and clearly states the background to the research question.
The methodology has been well structured and covers all areas. It defines the techniques and has stated that ethical approval was sought (include ethics reference number). However I think that primary and secondary outcomes should be more clearly defined. It is assumed that the primary outcome is operative time but this is not clearly described. I also have concerns over the description of the study design. The abstract describes it as a randomised study whereas the methodology (line 77) describes it as a prospective case control study. On line 83 it then reports that patients were randomised but does not clarify when, how and by whom. This area needs to be revised.
The results section is well presented and the appropriate statistical methodology has been used. The demographic table is well laid out and easy for the reader. The discussion/conclusion logically flows from the discussion and has used relevant contemporaneous references for discussion.
The included diagrams and images are useful.
Reviewer 3 Report
Manuscript ID: ijerph-1871379
Title: Safety of three-dimensional versus two-dimensional laparoscopic hysterectomy during the COVID-19 pandemic
Authors: Dariusz Kowalczyk, Szymon PiÄ…tkowski, Maja Porażko, Aleksandra Woskowska, Klaudia Szewczyk, Katarzyna Brudniak, Mariusz Wójtowicz, Karolina Kowalczyk
The manuscript by Dariusz Kowalczyk, Szymon PiÄ…tkowski, Maja Porażko, Aleksandra Woskowska, Klaudia Szewczyk, Katarzyna Brudniak, Mariusz Wójtowicz, Karolina Kowalczyk „Safety of three-dimensional versus two-dimensional laparoscopic hysterectomy during the COVID-19 pandemic” describe studies on the safety assessment of three-dimensional (3D) laparoscopy compared to two-dimensional (2D) laparoscopic hysterectomy during the COVID-19 pandemic.
The researches carried out are very interesting. The results are good described, but I suggest extending the information on the type of randomization that was used in the study?
Please standardize the font type (Table 2).
Are there any studies on the percentage of infection among healthcare professionals during surgical procedures? The study shows that the recoalescence time is the same, regardless of the method used. It can be assumed that the exposure of personnel to SARS-CoV-2 infection after surgery is the same. It would be worth adding information by what percentage the number of infected people decreased during the procedure using the laparoscopic technique in 3D imaging compared to 2D.
The authors conclude that "3D imaging improved the speed and reduced complications in laparoscopy [19]. A lower complication rate was also found in a study on three-dimensional laparoscopic cholecystectomy comparing to the two-dimensional method [20]". Is the 3D method more profitable from the economic point of view compared to the 2D method?
Round 2
Reviewer 2 Report
Thank you for your comments addressing my concerns. Thank you for clarifying the study as a prospective case-control study. I think it would be important to clearly define how the patients were assigned to the study groups as this is carries a potential for the introduction of bias and therefore would impact on the results of the study.
